# Diversity of lithophytic moss species in karst regions in response to elevation gradients

Yalin Jin ⓘ, Xiurong Wang*

College of Forestry, Guizhou University, Guiyang, Guizhou, China

* xrwang@gzu.edu.cn

**Data Availability Statement:** All minimal data files are available from the figshare database (DOI: 10.6084/m9.figshare.22430557.)

**Funding:** The author(s) received no specific funding for this work.

## Abstract

The distribution pattern of species diversity along various elevation gradients reflects the biological and ecological characteristics of species, distribution status and adaptability to the environment. Altitude, a comprehensive ecological factor, affects the spatial distribution of species diversity in plant communities by causing integrated changes in light, temperature, water and soil factors. In Guiyang City, we studied the species diversity of lithophytic mosses and the relationships between species and environmental factors. The results showed that: (1) There were 52 species of bryophytes in 26 genera and 13 families within the study area. The dominant families were *Brachytheciaceae*, *Hypnaceae* and *Thuidiaceae*. The dominant genera were *Brachythecium*, *Hypnum*, *Eurhynchium*, *Thuidium*, *Anomodon* and *Plagiomnium*; The dominant species were *Eurohypnum leptothallum*, *Brachythecium salebrosum*, *Brachythecium pendulum* etc. The number of family species and dominant family genera increased first and then decreased with the increase of altitude, and their distribution in elevation gradient III (1334-1515m) was the largest, with 8 families, 13 genera and 21 species. The elevation gradient I (970-1151m) was the least species distributed, with 5 families, 10 genera and 14 species. The dominant species with the largest number in each elevational gradient were *Eurohypnum leptothallum*, *Brachythecium pendulum*, *Brachythecium salebrosum* and *Entodon prorepens;* (2) There were five kinds of life forms in different elevation gradients, including Wefts, Turfs, Mat, Pendants and Tail. Among them, wefts and turfs appeared in all elevations, while a small amount of Pendants appeared in the area of elevational gradient I (970-1151m), and the most abundant life form was found in the range of elevational gradient III (1334-1515m); (3) Patrick richness index and Shannon-Wiener diversity index were highly significantly (p<0.01) positively correlated, both of which increased and then decreased with elevation, reaching a maximum at elevation gradient III (1334-1515m); The Simpson dominance index had a highly significant (p<0.01) negative correlation with the Patrick richness index and the Shannon-Wiener diversity index, which showed a decreasing and then increasing trend with increasing altitude; Pielou evenness index showed no discernible trend; (4) β diversity study revealed that while the similarity coefficient tended to decrease with increasing altitude, the species composition of bryophytes increased. The elevation gradient II (1151-1332m) and elevation gradient I (970-1151m) shared the most similarities, whereas elevation gradient III (1515-1694m) and elevation gradient I shared the least similarities (970-1151m). The findings can enrich the theory of the distribution pattern of lithophytic moss species diversity at distinct elevation

**Competing interests:** The authors have declared that no competing interests exist.

gradients in karst regions, and serve a scientific and reasonable reference for restoring rocky desertification and protecting biodiversity there.

## 1. Introduction

Research on biodiversity is crucial for the preservation of species [1]. Plant diversity research plays an important role in stabilizing natural ecosystems [2]. Low plant diversity can make vegetation system vulnerable to damage and affect the stability of ecosystem [3]. The higher the plant diversity, the higher the ecosystem stability [4]. Mosses are rich in species and second only to angiosperms among higher plants [5]. They are exceptionally drought-resistant and contribute significantly to the maintenance of biodiversity, water balance, allelopathic chemicals, soil and water conservation, ecological restoration and headwater conservation [6–11]. Lithophytic mosses, which swiftly absorb water and store a lot of it, are one of them. They formulate a biological microenvironment by releasing organic acids and other components that speed up the breakdown of the rock surface. This promotes the succession of vegetation, reduces the exposed area of the rock and improves the species diversity of lithophyte growing environment [12–17]. Therefore, understanding the distribution and status of lithophytic moss plants is important for the conservation of lithophyte species diversity and sustainable improvement of ecological context.

Altitude affects the spatial distribution of species diversity in plant communities as a whole by causing integrated changes in light, temperature, water and soil factors [18–20]. The relationship between plant species richness and altitude gradient is often hump-shaped, and the peak of plant diversity appears at the medium altitude, where the temperature and rainfall conditions are the best [21, 22]. Another common phenomenon is that species richness shows a monotonously decreasing trend with the increase of altitude, while there are few studies on species richness increasing monotonously with the increase of altitude or no obvious trend [23]. The pattern of species diversity distribution along various elevations reflects the biological and ecological traits of species, their distribution status and environmental suitability [24–26]. Elevation has a substantial impact on changes in moss species diversity, biomass, and water-holding capacity [27–29], The diversity of moss species increased significantly with altitude, especially at high altitudes [30]. The species diversity of lithophytic mosses in nature reserves, islands, lava flow, parks and steep mountains with wide variations in elevation gradients has received the majority of attention in previous research [31–38], but urban areas received less attention.

Guiyang is a typical karst city in the world's largest continuous karst belt with a potential rocky desertification and rocky desertification area that makes up 50.65% of the city's total land area [39]. The dominant families of moss plants in karst rocky desertification area are *Bryaceae*, *Pottiaceae*, *Brachytheciaceae*, *Hypnaceae*, *Thuidiaceae* and *Entodontaceae*, and the dominant species are *Bryum caespiticium*, *Eurohypnum leptothallum* and *Thuidium tamariscinum* etc [35, 36, 40, 41]. Current study on moss species diversity in karst areas focuses on the quantity and distribution of moss plant species in urban areas, urban walls and roadside slopes [42–44]. Huaxi district is where Guiyang city's highest and lowest elevation is found (970m-1697m above sea level). There are many mosses in this district, such as common ones as the *Eurhynchium* and *Pottia* [45], etc. Strengthening the study of lithophytic mosses in Guiyang City can provide scientific support for the management of rocky desertification. Little research has been reported on the relationship between species diversity of lithophytic mosses and elevation in karst areas, therefore, this paper selects the Huaxi district as the study area, and

studies the lithophytic moss communities at different elevation gradients in the region in order to explore the following questions:(1) Species composition, dominant family species and life type of bryophyte communities on distinct elevation gradients; (2) What is the pattern of moss species diversity distribution over the gradient of elevation? (3) What are the key factors affecting these distributions?

## 2. Study area and methods

### 2.1 Overview of the study area

Guiyang is situated in the midst of the mountain plain and hills, and is an important central city in southwest China, a typical karst area. Huaxi District of Guiyang City(26˚11'~26˚34'N, 106˚27'~106˚52'E) was chosen for this study, which has high topography in the southwest and low in the northeast, as well as a wide range of altitudes (970-1697m). It is a subtropical humid climate with an annual average temperature of 15.7˚C, annual precipitation of 1215.7mm, annual sunshine hours of 1162.6h and frost-free period of 339d. The main forest vegetation types include moist evergreen oak forests, mixed evergreen–deciduous forests and masson pine forests, etc [46].

### 2.2 Data sources and processing

The geographic elevation data (DEM) of the research region was obtained from Geospatial Data Cloud (http://www.gscloud.cn/), and using ArcGIS 10.7 independently to extract the study area's elevation. It could be seen that the Huaxi area's elevation was between 970 and 1697 meters, and was divided it into four gradients based on equal elevation intervals: I (970–1151 meters), II (1152–1333 meters), III (1334–1515 meters), and IIII (1516–1697 meters). Three representative community areas were selected in each elevation gradient to set up sample plots, 12 sample plots in total, which were numbered according to the elevation gradient (Fig 1 and Table 1). Two 10m×10m sample squares were selected for investigation in each sample plot, with a total of 24 sample squares, each of which was again divided into three middle sample squares (2m×2m) to collect lithophytic mosses. According to the community minimum area method, within each medium sample square, five small sample squares (10cm×10cm) were identified using metal frames according to the 5-point sampling method in order to sample and record their cover, habitat, life type, collection number and collection time, and a total of 360 samples were collected. The 360 samples were morphologically observed and identified. The methods were as follows: The morphology of the specimens was observed by the classical morphological method with the help of HWG-1 dissecting microscope and XSZ-107TS optical microscope, and the morphological observation and identification of the specimens were carried out with reference to "the Species Catalogue of China", "Volume 1 Plants", "Volumes 1–8 of Flora Bryophytorum Sinicorum" and "Volumes 1–3 of Bryophyte Flora of Guizhou China", etc. The specimens were stored in the bryophyte herbarium of the College of Forestry, Guizhou University. Statistics on the life types of bryophytes in different altitudinal gradients according to Magdefra's concept and distinction system [47].

### 2.3 Research methods

#### 2.3.1 α diversity index.

1. Patrick richness index indicates the richness of the plant community [48] and is calculated as follows:

$$D = S \tag{1}$$

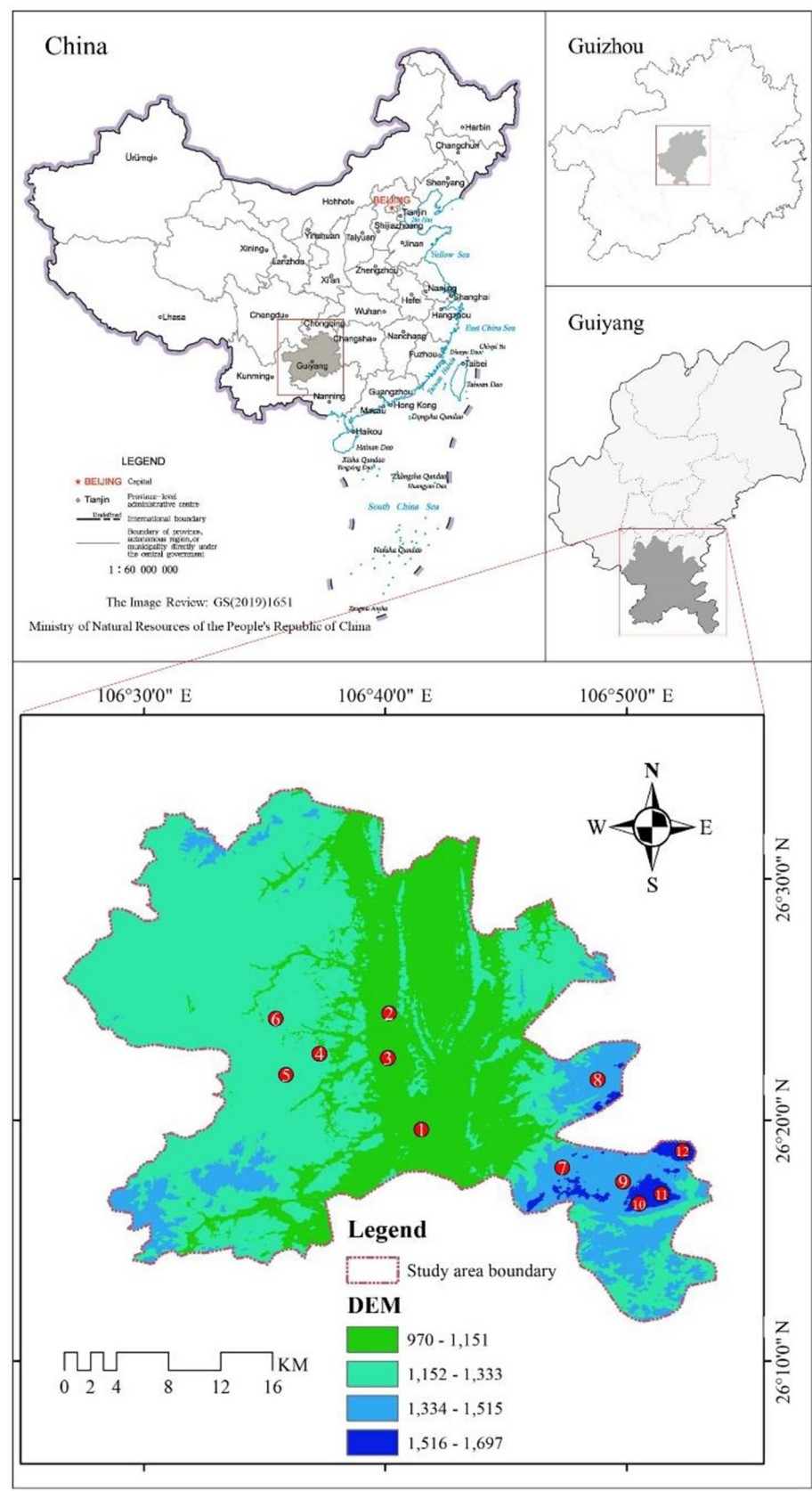

**Fig 1. Schematic diagram and plot layout of Huaxi district.**

**Table 1. Basic information of the sample site.**

| P-N | E | S-S | H | A | C-D (%) | V-C (%) | C-C(%) | S-C(%) | H-C(%) |
|-----|-----|-----|-----|-----|---------|---------|--------|--------|--------|
| I-1 | 1049.3 | Qingyan Ancient Town | Covered only with moss | Semi-sunny slope | 0.05 | 20 | 10 | 5 | 30 |
| I-2 | 1103.2 | Huaxi Park | Moss covered with herbs, shrubs and trees | Shady slope | 0.85 | 95 | 95 | 20 | 30 |
| I-3 | 1151.4 | Xishan Yujing Community | Moss covered with herbs, shrubs and trees | Semi-shady slope | 0.85 | 80 | 70 | 50 | 90 |
| II-1 | 1201.2 | Guizhou Normal University | Moss covered with herbs, shrubs and trees | Semi-sunny slope | 0.65 | 70 | 60 | 80 | 40 |
| II-2 | 1250.9 | Guizhou University of Finance and Economics | Moss covered with herbs, shrubs and trees | Semi-sunny slope | 0.5 | 60 | 40 | 30 | 80 |
| II-3 | 1302.4 | Tianhe Lake | Moss covered with herbs, shrubs and trees | Sunny slope | 0.9 | 90 | 85 | 20 | 90 |
| III-1 | 1351.3 | Qiantao Canyon | Moss covered with herbs, shrubs and trees | Sunny slope | 0.98 | 95 | 90 | 30 | 90 |
| III-2 | 1401.5 | Gusa Mountain Park | Covered only with moss | Semi-shady slope | 0.02 | 60 | 0 | 0 | 20 |
| III-3 | 1451.6 | Gao Po Ethnic Middle School | Mosses and shrubs and herbs | Shady slope | 0.66 | 95 | 45 | 70 | 80 |
| IIII-1 | 1516.6 | Yunding Skiing Grassland | Mosses and shrubs and herbs | Sunny slope | 0.2 | 20 | 0 | 0 | 60 |
| IIII-2 | 1551.8 | Pingzhai Village | Mosses and shrubs and herbs | Shady slope | 0.7 | 10 | 0 | 0 | 70 |
| IIII-3 | 1602.4 | Yunding Grassland | Mosses and shrubs and herbs | Semi-sunny slope | 0.05 | 70 | 0 | 0 | 80 |

Note: P-N. Plot number; E. Elevation; S-S. sample site; H. Habitat; A. Aspect; C-D. Canopy density; V-C. Vegetation coverage; C-C. Canopy coverage; S-C. Shrub coverage; H-C. Herb coverage

where $D$ is the diversity index, bryophyte species richness ($S$) = the number of species of lithophytic moss in different survey sample sites.

2. Shannon-Wiener diversity index is a measure of the community diversity and extent of heterogeneity at the species level, which integrates the sum of community species richness and evenness [49], and is calculated as follows:

$$H' = - \sum_{i=1}^{s} P_i \ ln \ P_i \tag{2}$$

$$Pi = Ni/N \tag{3}$$

Where $N$ is replaced by the total cover of mosses, $ni$ is replaced by the cover of the ith species, $H$ is the Shannon-Wiener diversity index, $Pi$ is the proportion of the ith species to the total.

3. Pielou evenness index reflects the relative density of individual species in the community [50] and is calculated as follows:

$$J = H/ln \ S \tag{4}$$

where $H$ is the Shannon-Wiener diversity index, $S$ is the number of species in the sample plot.

4. Simpson dominance index reflects the concentration of species distribution [51] and is calculated as follows:

$$\lambda = \Sigma pi^2 \tag{5}$$

where $\lambda$ is the Simpson dominance index, $Pi$ is the proportion of the ith species in the total.

**2.3.2 β diversity index.** β diversity index refers to the variability of species composition between different habitat communities along an environmental gradient or the turnover rate of species along an environmental gradient. The lower the number of common species among different communities or different points on an altitude gradient, the greater the β diversity [52]. The formula was calculated as follows:

$$\beta w = \frac{s}{ma} - 1 \tag{6}$$

Where $\beta w$ is β diversity index, $S$ is the total number of species in the studied system, $ma$ is the average number of species in each recipe or sample.

**2.3.3 Bryophyte importance value.** Importance value can indicate the roles and functions played by the species in the community, and the high importance value indicates the dominance of the species [53]. The formula is calculated as follows:

$$V = (C + F)/2 \tag{7}$$

where $V$ is the importance value, relative cover $C$ = (the cover of a bryophyte within sample site or sum of cover of all bryophytes in the sample site) × 100%, relative frequency $F$ = (frequency within a bryophyte sample site or sum of frequency of all bryophytes in the sample site) × 100%.

## 3. Results and analysis

### 3.1 Distribution and diversity characteristics of moss family species at different altitudes

There are 52 species of bryophytes in 26 genera and 13 families in the study area (Table 2). The number of family, genus and species gradually increased and then decreased as the altitude rose (Fig 2), and reached the maximum in elevation gradient III (1334-1515m), with 21 species in 8 families and 13 genera. The bryophyte families, genera and species below the altitude of 1152m were the least, with five families, ten genera, and fourteen species of moss.

The dominant families (Number of species contained ≥6 species) were *Brachytheciaceae* (4 genera and 17 species), *Hypnaceae* (5 genera and 9 species), and *Thuidiaceae* (3 genera and 6 species), with a total of 12 genera and 32 species in the three families, accounting for 61.54% of the total. Three families were distributed throughout the altitude study area (Fig 2), and the number of *Brachytheciaceae* and *Hypnaceae* increased and then decreased with increasing altitude, with the largest number of species distributed in elevation gradient III (1334-1515m), with 9 and 5 species, respectively. The distribution of *Thuidiaceae* was relatively uniform at all altitudes, and there were 2 species.

Six dominant genera (Number of species contained ≥3 species) were *Brachythecium*, *Hypnum*, *Eurhynchium*, *Thuidium*, *Anomodon*, and *Plagiomnium*, accounted for 51.93% of the total. The genus mainly distributed in elevation gradient III (1334-1515m) (Fig 3), and the distribution was the same as that of the dominant families. The number of species in the *Brachythecium*, *Eurhynchium*, *Thuidium* and *Plagiomnium* increased and then decreased with the increase of altitude, except for the *Anomodon*.

**Table 2. Species and genus proportion of bryophytes in Huaxi area.**

| Family name | Genus name | Species name | Genus (Percentage of total genus /%) | Species (percentage of total species /%) |
|---|---|---|---|---|
| *Hypnaceae* | *Eurohypnum, Homomalliu, Taxiphyllum, Pseudotaxiphyllum, Hypnum* | *E.leptothallum, H.yunnanense, H.plagiongium, T.cuspidifolium, P.densum, P.pohliaecarpum, H.calcicolum, H.hamulosum, H. cupressiforme* | 5(19.23) | 1(1.92), 2(3.85), 1 (1.92), 2(3.85), 3 (5.77) |
| *Pottiaceae* | *Hyophila, Didymodon, Tortella, Weissia* | *H.involuta, D.ditrichoides, T.tortuosa, W.exserta* | 4(15.38) | 1(1.92), 1(1.92), 1 (1.92), 1(1.92) |
| *Leucodontaceae* | *Pterogoniadelphus* | *P.esquirolii* | 1(3.85) | 1(1.92) |
| *Brachytheciaceae* | *Brachythecium, Eurhynchium, Homalothecium, Rhynchostegium* | *B.amnicolum, B.pendulum, B.garovaglioides, B.albicans, B. salebrosum, B.oedipodium, B.coreanum, B.populeum, B. helminthocladum, B.plumosum, B.kuroishicum, E.longirameum, E.angustirete, E. laxirete, H.leucodonticaule, R.ovalifolium* | 4(15.38) | 11(21.15), 3(5.77), 1 (1.92), 1(1.92) |
| *Thuidiaceae* | *Claopodium, Haplocladium, Thuidium* | *C.aciculums, H.angustifolium, H.microphyllum, T.kanedae, T. delicatulum, T.minutulum* | 3(11.54) | 1(1.92), 2(3.85), 3 (5.77) |
| *Anomodontaceae* | *Anomodon, Herpetineuron* | *A.perlingulatus, A.viticulosus, A.rugelii, H.toccoae* | 2(7.69) | 3(5.77), 1(1.92) |
| *Racopilaceae* | *Racopilum* | *R.cuspidigerum* | 1(3.85) | 1(1.92) |
| *Mniaceae* | *Plagiomnium* | *P.acutum, P.succulentum, P.maximoviczii* | 1(3.85) | 3(5.77) |
| *Meteoriaceae* | *Meteorium* | *M.subpolytrichum* | 1(3.85) | 1(1.92) |
| *Bryaceae* | *Bryum* | *B.dichotomum, B.calophyllum, B.pseudotriquetrum* | 1(3.85) | 3(5.77) |
| *Trachypodaceae* | *Trachypus* | *T.bicolor* | 1(3.85) | 1(1.92) |
| *Dicranaceae* | *Campylopus* | *C.gracilis, C.atrovirens* | 1(3.85) | 2(3.85) |
| *Entodontaceae* | *Entodon* | *E.prorepens* | 1(3.85) | 1(1.92) |

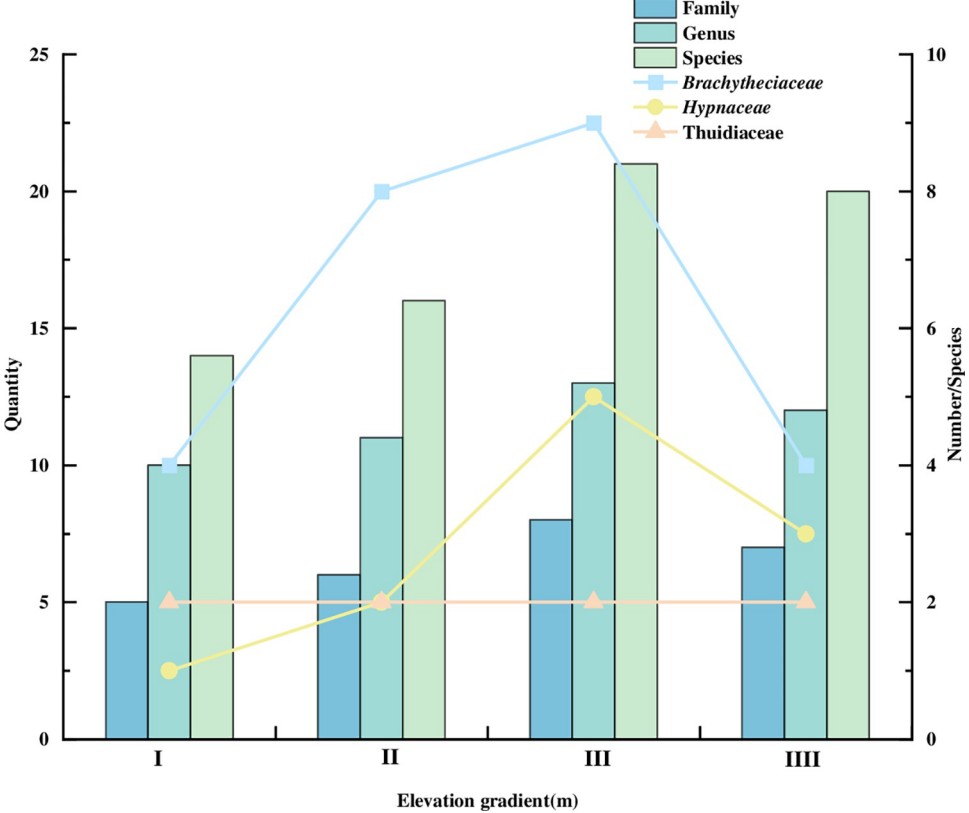

**Fig 2. Distribution of species and dominant families at different elevation gradients.**

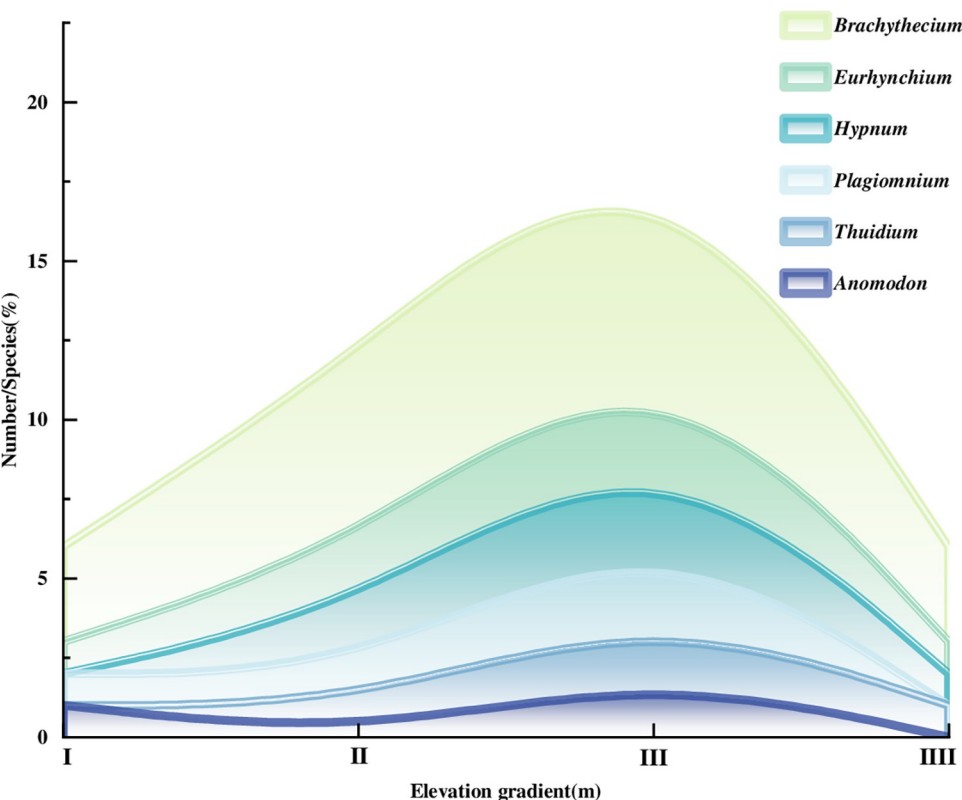

**Fig 3. Vertical distribution of dominant bryophyte genera.**

### 3.2 Important value of bryophyte communities at different altitudes

Among the 52 bryophytes, the top ten ranked by importance value were shown in Table 3. the largest importance value of 0.1906 (p<0.01) was found for *Eurohypnum leptothallum*, which was highly significant different from that of other bryophytes, and its frequency (18.33%) and cover (19.79%) were also the largest. The distribution of dominant species at different altitudes was shown in Fig 4. It could be seen that the *Eurohypnum leptothallum* was the main dominant species, reaching its highest and second highest peaks at elevation gradient I (970-1151m) and elevation gradient III (1334-1515m), respectively. The dominant species with the highest frequency in each elevation gradient were *Eurohypnum leptothallum*, *Brachythecium pendulum*, *Brachythecium salebrosum* and *Entodon prorepens*. Among them, the number of *Entodon prorepens* gradually increased with the elevation, while the frequency of other bryophytes was less distributed in the elevation IIII section.

### 3.3 Life form composition of moss plants at different altitudes

There were five life forms at different elevation gradients, such as wefts, turfs, mat, pendants and tail, among which the average percentage of Wefts were 73%, for example, *Eurohypnum leptothallum*, *Brachythecium amnicolum*, etc, followed by turfs, the average percentage were 15%, for example, *Hyophila involuta*, *Didymodon ditrichoides*, etc. Mat, Pendants and Tail were the least, with an average of 6%, 4% and 2%, respectively, such as *Plagiomnium succulentum*, *Pterogoniadelphus esquirolii* and *Herpetineuron toccoae*. The life forms of different elevation gradients also differed (Fig 5), with wefts and turfs occurred in all elevation gradients, but pendants occurred only in small amounts in the range of elevation gradient I (970-1151m),

**Table 3. Ranking of important values of bryophytes.**

| serial number | Genus name | relative frequency | relative coverage | importance value |
|---|---|---|---|---|
| 1 | *Eurohypnum leptothallum* | 18.33% | 19.79% | 0.1906 |
| 2 | *Brachythecium salebrosum* | 6.67% | 7.61% | 0.0714 |
| 3 | *Brachythecium pendulum* | 5.83% | 5.26% | 0.0555 |
| 4 | *Brachythecium amnicolum* | 5.00% | 5.59% | 0.0530 |
| 5 | *Hyophila involuta* | 5.56% | 4.98% | 0.0527 |
| 6 | *Entodon prorepens* | 5.56% | 4.69% | 0.0512 |
| 7 | *Brachythecium garovaglioides* | 3.61% | 4.35% | 0.0398 |
| 8 | *Thuidium delicatulum* | 3.33% | 3.65% | 0.0349 |
| 9 | *Brachythecium albicans* | 3.06% | 2.67% | 0.0286 |
| 10 | *Eurhynchium longirameum* | 3.06% | 2.61% | 0.0283 |

while at elevation gradient III (1334-1515m), the life types were the most abundant, including wefts, turfs, mat, pendants, and tail.

## 3.4 Characteristics of bryophyte diversity at different elevation gradients

**3.4.1 α diversity and its differences across elevation gradients.** The α diversity index of bryophytes at various elevation gradients were shown in Fig 6. The trends of Patrick richness index and Shannon-Wiener diversity index were consistent, showing an increasing and then decreasing trend with elevation, as follows: elevation gradient III (33 species, 2.53) > elevation gradient IIII (29 species, 2.52) > elevation gradient II (25 species, 2.4) > elevation gradient I

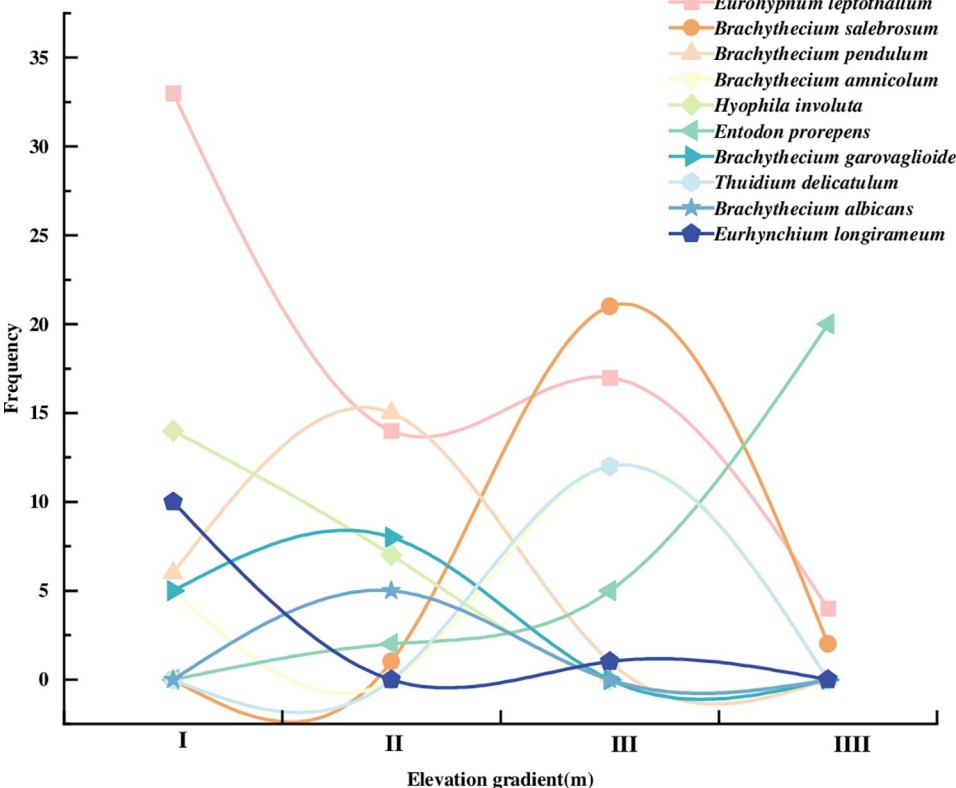

**Fig 4. Vertical distribution of dominant bryophyte species.**

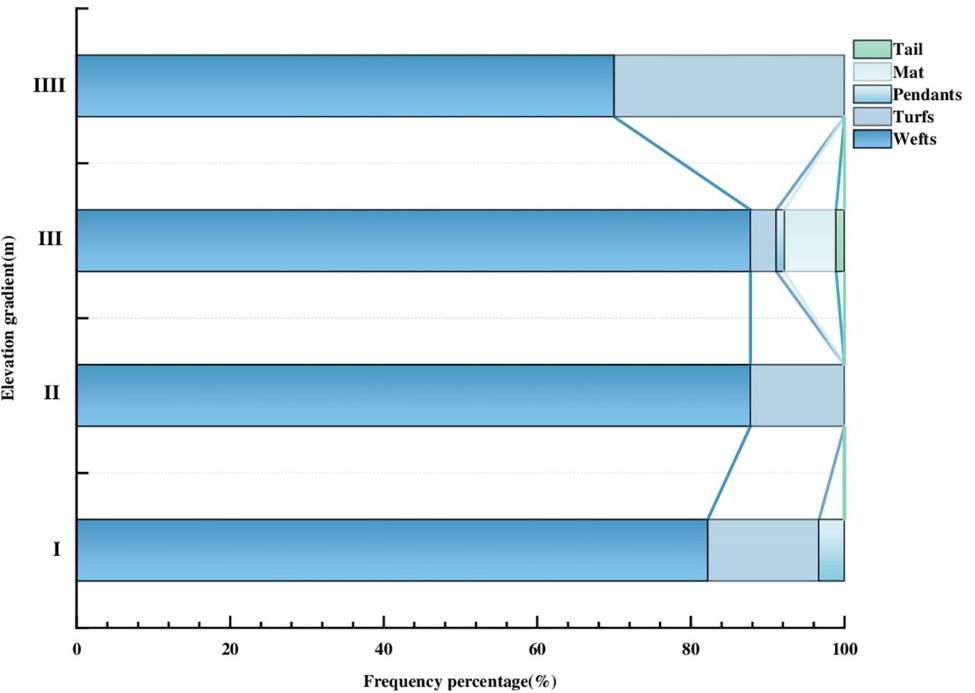

**Fig 5. Life types of species at different altitudes.**

(22 species, 2.13). The trend of Simpson dominance index was opposite to the above two, that was, it first decreased and then increased with the elevation, specifically as follows: Elevation gradient I (0.18) > Elevation gradient II (0.13) > Elevation gradient IIIII (0.11) > Elevation gradient III (0.09). The variation pattern of Pielou evenness index was: elevation gradient II (0.87) > elevation gradient III (0.81) > elevation gradient III and elevation gradient I (0.75). There were significant differences ($p<0.05$) between the richness, diversity and dominance indices of elevation gradient III (1334-1515m) and the other three gradients.

In summary, Patrick richness index was lowest at elevation gradient I (970-1151m) and increased with the elevation. The reason might be that at elevation gradient I (970–1151 m), the habitat of bryophytes had low canopy density because there were no mixed tree forests, low shrubs, or herbaceous plants. However, as elevation increased, the air humidity and canopy density increased significantly, Patrick richness index also increased. The differences of Pielou evenness index were not significant, indicating that different elevation gradient had little effect on the individual evenness of the species.

The correlation between α diversity index and environmental factors showed (Fig 7) that both Patrick richness index and Shannon-Wiener diversity index had highly significant ($p<0.01$) positive correlations with altitude and air humidity, and highly significant ($p<0.01$) negative correlations with air temperature. There was a highly significant ($p<0.01$) positive correlation between Simpson dominance index and air temperature with a correlation coefficient of 0.898, and a highly significant ($p<0.01$) negative correlation with altitude and air humidity with correlation coefficients of -0.799 and -0.792, respectively. Correlations between environmental factors of different elevation gradients showed a highly significant ($p<0.01$) positive correlation between altitude and air humidity and a highly significant ($p<0.01$) negative correlation with air temperature. There was a highly significant ($p<0.01$) positive correlation between slope orientation and light intensity, and a highly significant ($p<0.01$) negative correlation between canopy density and tree cover. It could be seen that the environmental

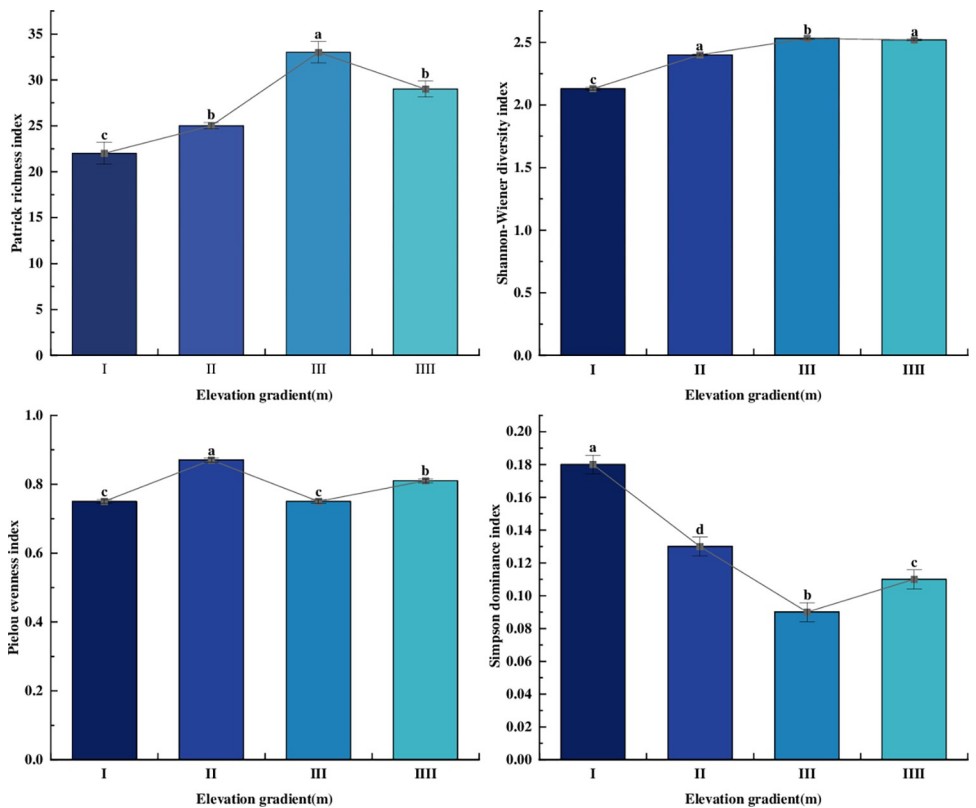

**Fig 6. Effects of altitude α diversity index.**

factors affecting species diversity were mainly elevation, air humidity, air temperature, and slope direction affected the intensity of light, which led to the difference of air humidity and temperature.

**3.4.2 Trends in β diversity index at different elevation gradients.** The similarity of species composition of moss communities across the elevation gradient was showed in Table 4. With the altitude increased, the similarity coefficient decreased and the difference of bryophyte species composition increased (11.76%-40%). Elevation gradients II (1152–1333 m) and I (970–1151 m) shared the highest similarity of 40%, indicating that they shared a large number of the same moss species. Elevation gradient IIII (1516-1697m) and elevation gradient I (970-1151m) had the lowest similarity coefficient of 11.76%. The reason might be that the air humidity increased as the altitude rose, and the air temperature and the temperature of moss-covered rocks decreased, which was conducive to the growth and variety of moss. However, above elevation gradient III (1334-1515m), the differences of moss species began to increase. The possible reason was that the sample land on gradient IIII(1516–1697) were located in artificial scenic areas, such as Genting Grassland, which had low canopy coverage, almost no tree and only a few shrubs. As a result, the moss plants lacked nutrient elements and favorable environmental conditions for growth, so the species decreased and the similarity coefficient reached the lowest.

## 4. Discussion

### 4.1 Effects of different elevation gradients on the diversity of moss species

As an important component of biodiversity, species diversity primarily reflected the abundance of biological resources, and an increase in species diversity would enhance the stability

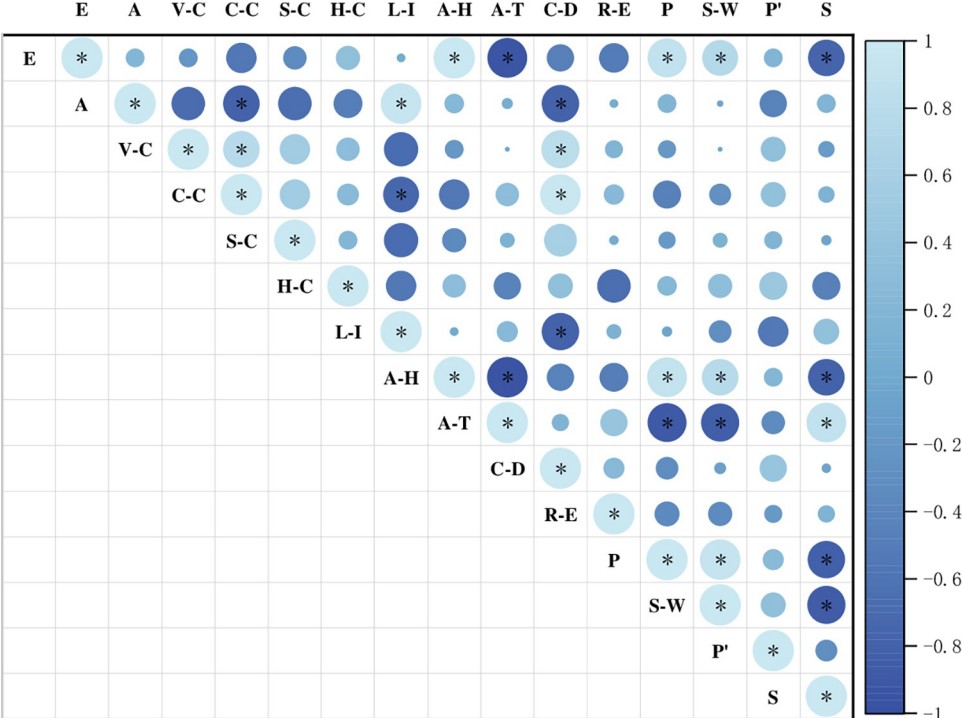

**Fig 7. Pearson correlation analysis of α diversity index and environmental factors.** Note;* p < 0.05**;P <0.01;E. Elevation;A.Aspect;V-C.Vegetation coverage; C-C. Canopy coverage; S-C. Shrub coverage; H-C. Herb coverage; L-I. Light intensity; A-H. Air humidity; A-T. Air temperature; C-D. Canopy density; R-E. Rock exposure; P. Patrick richness index; S-W. Shannon-Wiener diversity index; P'. Pielou evenness index; S. Simpson dominance index.

and productivity of a community or ecosystem [54]. Previous research had demonstrated that the effects of elevation gradient on variables like temperature, light, soil, and moisture had a significant impact on the composition of species diversity [55, 56].

Firstly, elevation gradient significantly affected the composition and quantity of family and genus [57]. In the elevation gradient of 1400-2200m in the Jiufeng Mountain Nature Reserve, the percentage of bryophyte species increased with the elevation [58]. The number of bryophyte genera and species in Pingding Mountain, the main peak of the Lesser Khingan Mountains, showed two peaks within the elevation gradient 304-1429m [59]. The bryophyte species in the Daba Mountain National Nature Reserve showed a single peak value with the increase of altitude [60].By studying the distribution of lithophytic moss and the influence of environmental factors on it at different altitude gradients, we found that the number of species, genera and family increased first and then decreased with elevation, which were in line with those of earlier studies [32]. Most habitats in elevation gradient III (1334-1515m) were forest stony habitats with large canopy coverage and low human disturbance, and there were very

**Table 4. Differences of β diversity index at different elevation gradients (%).**

| Elevational gradient (m) | I | II | III | IIII |
|---|---|---|---|---|
| I | 100 | 40.00 | 17.14 | 11.76 |
| II | | 100 | 21.62 | 22.22 |
| III | | | 100 | 19.51 |
| IIII | | | | 100 |

abundant moss plant species. In contrast, in elevation gradient IIII (1516-1697m), the increase in light and radiation were not favorable for the growth of moss plants [61], so the number of family, genera and species was low. The innovation of this study was that we selected karst urban areas, where the habitat conditions changed greatly, with typical karst mountain features. However, the previous study objects were mostly bryophytes distributed in the outskirts of the mountain, belonging to the subalpine cold and temperate forests with rich environmental vegetation and large canopy [32]. Our results compensated for the diversity of moss in special habitats and provided a scientific basis for the control of rocky desertification.

Secondly, altitude altered the composition of life form in proportion and impacted its pattern of distribution [62–64]. The Alpine area had a large altitude gradient, and its bryophyte life forms are related in different altitude gradients, namely plagiotropic, cushion-turf, dendroid and thallose forms [65]. Bryophyte species on different substrates were studied along the altitude gradients in Canary Islands. The results showed that bryophyte life forms were most abundant in the higher altitude gradients [66]. Studies on bryophyte life forms under different altitudinal gradients in the southern Appalachian Mountains showed that altitude had a significant effect on bryophyte life forms. Weft, threadlike, tall turf and short turf life forms were strongly correlated with high altitude sites [67]. This study carried out the effect of elevation gradient on life form. It was discovered that there were five different life forms in the study area, including wefts, turfs, mat, pendants and tail, which were gradually enrich as the elevation increased and most abundant in elevation gradient III (1332-1513m). Bryophytes of different life form could reveal the link between plants and the environment and reflect the vertical structure of the community. Their distribution patterns on the altitude gradient could give data to forecast changes in species diversity under climate change [68–70].Wefts and Turfs occurred in all elevation gradients, which might be due to the fact that both of them could grow on the rock surface, which was consistent with the research results of Zeke Li et al. [71].

At last, elevation gradient was an important environmental factor affecting species α diversity and β diversity [72]. The α diversity index reflected the compositional characteristics, abundance and diversity of species within the community [73–75]. According to earlier research, the α diversity index of bryophytes showed five trends with increasing elevation, such as increasing and then decreasing(bimodal curve), first decreasing followed by increasing, monotonically increasing, monotonically decreasing, increasing followed by decreasing [58, 59, 76–78]. This study found that the Patrick richness and Shannon-Wiener diversity index increased first and then decreased with the increase of altitude. The reason might be that the bryophytes in elevation gradient I (970-1151m) were significantly disturbed by humans, which had an impact on their growth. Additionally, most of the sample sites were exposed habitats that suffered from sunlight exposure, which resulted in fewer bryophytes. With increasing elevation, the habitats of elevation gradient II (1152-1333m) and III (1332-1513m) were covered with a large amount of vegetation, air humidity, canopy coverage and so abundance also showed up as an increase. However, the light intensity at elevation gradient IIII (1516-1697m) increased, which was not conducive to the growth of moss. The β diversity index reflected the response of species composition and diversity among communities to environmental gradients [79]. The current studies on β diversity indices of bryophytes at various altitudinal gradients had mainly explored the altitudinal intervals with the fastest succession rates of bryophytes [80]. There was some uncertainty on the trend of the β diversity index along altitude. In this study, we found that increasing altitude contributed to the decrease of β diversity index of bryophytes and the increase of species composition differences and succession rate. The distribution pattern of moss species diversity along the elevation gradient of Guiyang city was first increasing and then decreasing (single-peaked curve), which was consistent with the results of

one of the previous studies. This study intends to further investigate the effect of pollution on mosses distribution in urban areas based on further data collection.

## 4.2 Limitations and outlook

The study on the diversity of moss species at various elevation gradients in karst areas will clarify the relationship between species diversity and elevation, provide a theoretical source for understanding conditions of species to the altitude and a scientific basis for the conservation and management of rocky desertification, so as to strengthen the conservation of biodiversity in karst areas accordingly. However, this study still had the following limitations. First of all, in this study, portable instruments were used to record environmental factors such as air temperature and air humidity at the sample sites during field research, and only transient data were collected. However, air temperature and humidity are dynamic. Future recording of the dynamic processes of environmental factors will provide a clearer understanding of the factors affecting species diversity at elevation. Secondly, rainfall, solar radiation and canopy action also have a strong influence on abundance [81–83], for example, high temperatures at low elevations can be cooled by high rainfall, and high solar radiation at high elevations can be mitigated by canopy. Although there were changes in light caused by solar radiation and depression from the action of the canopy, the influence of precipitation was not taken into account in the study area, and the specific internal factors affecting richness were not examined in greater detail. Future research taking into account rainfall and other factors will provide stronger evidence for changes in species diversity caused by elevation. Of course, bryophyte distribution in urban areas is not only affected by altitude, but also by heavy metal pollution, human disturbance, urban management, etc [84].

## 5. Conclusion

1. There were 13 families, 26 genera, and 52 species of lithophytic mosses in different altitude gradients in Guiyang City. The number of family, genera, species and dominant family and genera gradually increased and then decreased with elevation, reached the maximum in gradient III (1334-1515m). The dominant families were *Brachytheciaceae*, *Hypnaceae* and *Thuidiaceae*, and the dominant genera were *Brachythecium*, *Hypnum*, *Eurhynchium*, *Thuidium*, *Anomodon* and *Plagiomnium*.

2. There were five different life forms: wefts, turfs, mat, pendants and tail. Wefts and Turfs could be found in all gradients of altitude. Life type gradually became more diverse as altitude rises, and they were the most abundant in gradient altitude III (1334-1515m). There were only a small amount of pendants appeared at elevation gradient I(970-1151m).

3. Both Patrick richness and Shannon-Wiener diversity index showed a trend of increasing and then decreasing with the altitude, reached a maximum at elevation gradient III (1334-1515m) and a minimum at elevation gradient I (1334-1515m). The change of Simpson dominance index was generally opposite to the change of the above two indices, showing a decreasing trend followed by an increasing trend, and the difference of Pielou uniformity index was not significant.

4. The distribution pattern of moss species diversity along the elevation gradient in Guiyang City was first increasing and then decreasing (single-peaked curve).

## Author Contributions

**Data curation:** Yalin Jin.

**Formal analysis:** Yalin Jin.

**Funding acquisition:** Xiurong Wang.

**Investigation:** Yalin Jin.

**Methodology:** Yalin Jin.

**Project administration:** Yalin Jin.

**Software:** Yalin Jin.

**Writing – original draft:** Yalin Jin.

**Writing – review & editing:** Xiurong Wang.

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
