## [Decision Letter · Decision Letter 0]

21 Feb 2023

PONE-D-22-35414Diversity of lithophytic moss species in karst regions in response to elevation gradientsPLOS ONE

Dear Dr. Jin,

Thank you for submitting your manuscript to PLOS ONE. After careful consideration, we feel that it has merit but does not fully meet PLOS ONE’s publication criteria as it currently stands. Therefore, we invite you to submit a revised version of the manuscript that addresses the points raised during the review process.

We look forward to receiving your revised manuscript.

Kind regards,

Ying Ma, Ph.D.

Academic Editor

PLOS ONE

Journal Requirements:

"The authors are grateful for the financial support from National Natural Science Foundation of China (NSFC) (31960328)."

7. We note that Figure 1 in your submission contain [map/satellite] images which may be copyrighted. All PLOS content is published under the Creative Commons Attribution License (CC BY 4.0), which means that the manuscript, images, and Supporting Information files will be freely available online, and any third party is permitted to access, download, copy, distribute, and use these materials in any way, even commercially, with proper attribution. For these reasons, we cannot publish previously copyrighted maps or satellite images created using proprietary data, such as Google software (Google Maps, Street View, and Earth). For more information, see our copyright guidelines: http://journals.plos.org/plosone/s/licenses-and-copyright.

Reviewers' comments:

Reviewer's Responses to Questions

**Comments to the Author**

1. Is the manuscript technically sound, and do the data support the conclusions?

Reviewer #1: No

Reviewer #2: Yes

2. Has the statistical analysis been performed appropriately and rigorously? 

Reviewer #1: Yes

Reviewer #2: Yes

3. Have the authors made all data underlying the findings in their manuscript fully available?

Reviewer #1: Yes

Reviewer #2: Yes

4. Is the manuscript presented in an intelligible fashion and written in standard English?

Reviewer #1: Yes

Reviewer #2: Yes

5. Review Comments to the Author

Reviewer #1: Dear Editor,

I has now commented on manuscript, entitled “Diversity of lithophytic moss species in karst regions in response to

elevation gradients”.

The paper falls within the general scope of the journal. The title reflects the content, and it is recommended that keywords are not in the title. Please, in the introduction more effort should be put into explaining the main reasons for the study.

In general the description of methods is good.

The discussion is good? And what is the future perspective of this research?

Language overall is good for non-native speakers, but the writing could be improved with a quick revision by a native English speaker.

I would recommend minor revision for the paper for it to be considered further for publication.

Sincerely

Reviewer #2: The work is quite well written, it is interesting and it gives a contribution on the altitudinal distribution of bryophytes, however I would like to make some criticisms: 1) the altitudinal gradient is not very marked, just over 700 meters of altitudinal range, although the authors manage to highlight significant differences on the gradient. 2) another aspect to be developed a little more is the relationship and the influence of the urban environment on the muscinal species 3) the references should in many cases be strengthened and I have reported it in the attached pdf

6. PLOS authors have the option to publish the peer review history of their article (what does this mean?). If published, this will include your full peer review and any attached files.

Reviewer #1: No

Reviewer #2: No

---

## [Author Response · Author response to Decision Letter 0]

16 May 2023

Dear reviewers:

Thank you very much for your suggestion of modification. Each of these points responds as follows:

Questions about Financial Disclosure: Funded by National Natural Science Foundation of China (NSFC) (31960328)( https://www.nsfc.gov.cn/publish/portal0/default.htm). The funders had no role in study design, data collection and analysis, decision to publish, or preparation of the manuscript.

Questions about Data Availability statements: All minimal data files are available from the figshare database (DOI: 10.6084/m9.figshare.22430557.)

Ethics Statement: N/A.

The methods section does not require permission, as a brief statement:The research methods are cited in the literature and no permission is required.

Copyright required for Figure 1: The map in Figure 1 is available for public. download by relevant departments in China.

Sincerely looking forward to your reply

Yalin Jin

April 5th, 2023

---

## [Decision Letter · Decision Letter 1]

22 May 2023

Diversity of lithophytic moss species in karst regions in response to elevation gradients

PONE-D-22-35414R1

Dear Dr. Jin,

We’re pleased to inform you that your manuscript has been judged scientifically suitable for publication and will be formally accepted for publication once it meets all outstanding technical requirements.

Kind regards,

Ying Ma, Ph.D.

Academic Editor

PLOS ONE

Additional Editor Comments (optional):

Reviewers' comments:

Reviewer's Responses to Questions

**Comments to the Author**

1. If the authors have adequately addressed your comments raised in a previous round of review and you feel that this manuscript is now acceptable for publication, you may indicate that here to bypass the “Comments to the Author” section, enter your conflict of interest statement in the “Confidential to Editor” section, and submit your "Accept" recommendation.

Reviewer #1: All comments have been addressed

Reviewer #2: All comments have been addressed

2. Is the manuscript technically sound, and do the data support the conclusions?

Reviewer #1: Yes

Reviewer #2: Yes

3. Has the statistical analysis been performed appropriately and rigorously? 

Reviewer #1: Yes

Reviewer #2: Yes

4. Have the authors made all data underlying the findings in their manuscript fully available?

Reviewer #1: Yes

Reviewer #2: Yes

5. Is the manuscript presented in an intelligible fashion and written in standard English?

Reviewer #1: Yes

Reviewer #2: Yes

6. Review Comments to the Author

Reviewer #1: (No Response)

Reviewer #2: I appreciate the work done by the authors who have improved the manuscript according to the indications of the reviewers, increasing the bibliographic references and developing the missing parts, now we have a better discussion. On this second version I point out that the name of a family of mosses which appears 15 times always in the Latin singular form "Brachytheciacea" must be corrected; it must be corrected with an extra e at the end : "Brachytheciaceae". I missed this detail in my first review. Moreover, in the last sentence at the end of the Discussion it would be appropriate to specify that the authors, on the basis of further data to be collected, intend to deeply investigate the effects of pollution in the urban area on the distribution of mosses.

7. PLOS authors have the option to publish the peer review history of their article (what does this mean?). If published, this will include your full peer review and any attached files.

Reviewer #1: No

Reviewer #2: No

---

## [Editor Report · Acceptance letter]

21 Jun 2023

PONE-D-22-35414R1 

Diversity of lithophytic moss species in karst regions in response to elevation gradients 

Dear Dr. Jin:

I'm pleased to inform you that your manuscript has been deemed suitable for publication in PLOS ONE. Congratulations! Your manuscript is now with our production department. 

Kind regards, 

on behalf of

Dr. Ying Ma 

Academic Editor

PLOS ONE